# Active Instruction Tuning: Improving Cross-Task Generalization by Training on Prompt Sensitive Tasks

**Po-Nien Kung,   Fan Yin,   Di Wu,   Kai-Wei Chang,   Nanyun Peng**

University of California, Los Angeles

{ponienkung,fanyin20,diwu,kwchang,violetpeng}@cs.ucla.edu

## Abstract

Instruction tuning (IT) achieves impressive zero-shot generalization results by training large language models (LLMs) on a massive amount of diverse tasks with instructions. However, how to select new tasks to improve the performance and generalizability of IT models remains an open question. Training on all existing tasks is impractical due to prohibiting computation requirements, and randomly selecting tasks can lead to suboptimal performance. In this work, we propose *active instruction tuning* based on prompt uncertainty, a novel framework to identify informative tasks, and then actively tune the models on the selected tasks. We represent the informativeness of new tasks with the disagreement of the current model outputs over perturbed prompts. Our experiments on NIV2 and Self-Instruct datasets demonstrate that our method consistently outperforms other baseline strategies for task selection, achieving better out-of-distribution generalization with fewer training tasks. Additionally, we introduce a task map that categorizes and diagnoses tasks based on prompt uncertainty and prediction probability. We discover that training on ambiguous (prompt-uncertain) tasks improves generalization while training on difficult (prompt-certain and low-probability) tasks offers no benefit, underscoring the importance of task selection for instruction tuning.[1]

## 1 Introduction

Recently, instruction tuning has shown great success in improving large language models' cross-task generalizability. When training large language models (LLM) with a wide range of tasks with instructions, models like T0 (Sanh et al., 2021), FLAN (Wei et al., 2021), TK-Instruct (Wang et al., 2022b), Instruct-GPT (Ouyang et al., 2022), Alpaca (Taori et al., 2023) and Vicuna (Chiang et al.,

---

[1] Our code and data can be found at https://github.com/PlusLabNLP/Active-IT

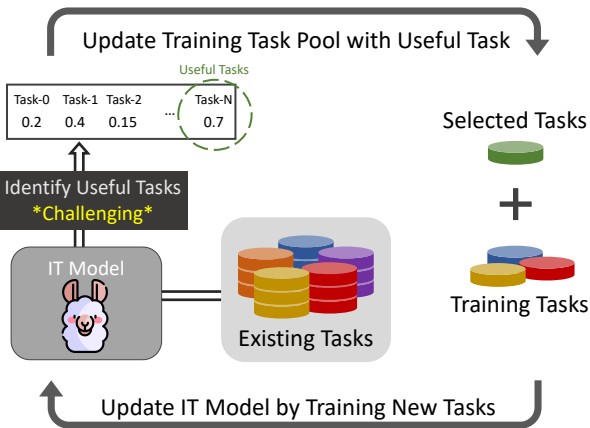

Figure 1: Our proposed Active Instruction Tuning framework. Given an instruction tuning (IT) model with a sizeable existing task pool, we actively **identify and select** useful tasks from it and add them to the training task pool. After that, we train a new IT model with the updated training task pool. By continuing this loop, we expect to improve the IT model's cross-task generalization ability efficiently. The main challenge lies in **identifying useful tasks**, for which we propose to select prompt-sensitive tasks.

2023) can perform well on unseen task. The performance can be further boosted by increasing the number of diverse training tasks (Xu et al., 2022; Wang et al., 2022b; Longpre et al., 2023; Chung et al., 2022). Based on this observation, many recent studies scale up instruction-tuning datasets by manually or automatically curating more tasks with instructions. For example, T0 and FLAN have 60 tasks. The NIV2 (Wang et al., 2022b) benchmark extends its dataset to over 800 English training tasks. Self-Instruct (Wang et al., 2022a) and Unnatural Instructions (Honovich et al., 2022) prompt LLMs to generate over 50K instruction tuning data, and recently, Dynosaur (Yin et al., 2023a) dynamically curates over 80K instruction tuning data from Huggingface datasets (Lhoest et al., 2021), which is still continuously expanding.

However, as the scale of datasets grows rapidly, it becomes impractical to train on all existing tasks due to overwhelming computing costs. One naive solution is to randomly sample tasks for training,

but it can potentially select less informative tasks, leading to suboptimal results (Wang et al., 2023). Therefore, it is crucial to employ an efficient task selection strategy that identifies the most novel and informative tasks for instruction tuning.

Data selection has been explored under active learning and multi-task learning frameworks. Despite its prevalence, we argue that they are not applicable to task selection for instruction tuning. Specifically, active learning methods have focused on selecting the most useful *instances* for a single task, using either uncertainty-based intuitions such as entropy (Settles, 2009), Monte Carlo dropout (Gal and Ghahramani, 2016), or ensemble disagreement (Houlsby et al., 2011; Siddhant and Lipton, 2018). However, these uncertainty measurements can only measure uncertainty at *instance-level*, and will become less effective when applied to *task-level* selections as the scales of uncertainty values are not comparable across tasks. In multi-task learning, previous research (Ivison et al., 2022; Poth et al., 2021; Kung et al., 2021) has explored measuring task usefulness by assessing its similarity to the target task. While these methods can enhance performance when aware of the target tasks, they are not suitable for instruction tuning, which aims to improve *overall generalization* to arbitrary *unseen* tasks.

In this work, we introduce **Active Instruction Tuning**, a framework that aims to actively identify informative new tasks for an IT model to continuously improve its cross-task generalization ability (refer to Figure 1). While being related to active learning, our task is more challenging. Unlike active learning, which focuses on improving **performance on a single target task** by identifying useful *instances*, our goal is to identify tasks that enhance **overall generalization**, a novel concept not explored in previous AL research.

To identify informative new tasks for Active Instruction Tuning, we propose *prompt uncertainty* (refer to Figure 2), a novel task-level uncertainty metric that measures the sensitivity of an IT model against instruction perturbations for a task. Specifically, with a task instruction and a few unlabeled instances, we assess the disagreement of model predictions against original and perturbed prompts on multiple instances to obtain an average disagreement score. We then select and train the model with the most prompt-uncertain tasks to enhance the overall cross-task generalization ability. Since

this uncertainty method does not require labeled instances for a task, we can also apply prompt uncertainty to determine novel tasks to manually annotate if needed.

We further explore using *prompt uncertainty* to understand task characteristics and diagnose potential issues. Motivated by Data Map (Swayamdipta et al., 2020), which utilizes instance-level training dynamics to categorize and diagnose data quality, we propose **Task Map**, the first task diagnosing method that categorizes tasks based on Prompt Uncertainty and Prediction Probability. Based on the Task Map, we categorize tasks into **Ambiguous**, **Easy** and **Difficult**, inspired by prior in-context learning research (Xie et al., 2021; Pan et al., 2023) to facilitate analysis.

We conduct experiments on two instruction tuning setting: TK-Instruct models with NIV2 dataset, which generalize to unseen tasks, and Alpaca models with Self-Instruct dataset, which generalize to unseen instructions, following our categorization in Kung and Peng (2023). Results show that our active instruction tuning method consistently outperforms baseline methods (random sampling, generation perplexity) for both instruction tuning setting, demonstrating the effectiveness of our approach. Moreover, we discover that while instruction tuning with **Ambiguous** tasks can improve generalization effectively, **Difficult** tasks offers no benefit, underscoring the importance of task selection in instruction tuning. Our contributions can be summarized as follows:

- We introduce Active Instruction Tuning, a framework to efficiently improve the IT model's generalization ability in large-scale instruction tuning.

- We propose Prompt Uncertainty, a task-level uncertainty measurement for IT, which can identify novel/informative tasks to improve IT models' zero-shot generalization.

- We further propose Task Map, a task diagnosing tool that categorizes tasks based on their prompt uncertainty and prediction probability, providing insights into task characteristics and quality.

## 2 Method

### 2.1 Active Instruction Tuning

The Active Instruction Tuning framework is illustrated in Figure 1. In reality, when the number of

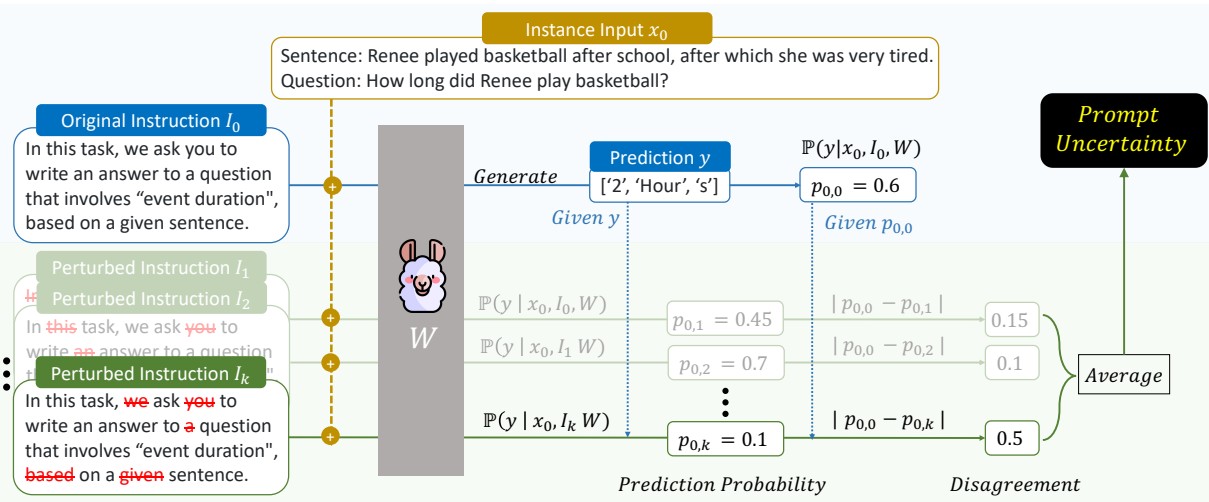

Figure 2: We demonstrate how we measure the *prompt uncertainty* of a model $W$ to a task. Given the original instruction $I_0$ and instance input $x_0$ of a task to the model, we first get prediction $y$ and its sentence probability $p_{0,0}$. Next, we randomly drop words (highlighted in red) from the original instructions to create $k$ perturbed instructions. We measure the model's prediction probability of $y$ given each of the perturbed instructions and input $x_0$. Finally, we calculate the average absolute difference (disagreement) between the prediction probability of using original and perturbed instructions, providing an estimate of the model's prompt uncertainty for $x_0$. We can further aggregate this prompt uncertainty scores across $n$ instances for a task. Further details can be found in section 2.

tasks is large and continuously expanding, training on all existing tasks becomes impractical due to the overwhelming computing cost. To efficiently improve an instruction tuning model, we can apply a task selection method to actively select the tasks that benefit the current model the most. By repeating this model training and task selection loop, we can continually improve instruction-tuned models' generalization to unseen tasks.

For the experiment, we use a large training task pool of fixed size. The training procedure consists of multiple iterations. In the first iteration, a small number of tasks are randomly sampled to train a weak instruction-tuned model. In subsequent iterations, we actively select the most useful tasks based on the previous model and train a new model with the selected tasks. We evaluate different task selection strategies by testing the model on unseen tasks at each iteration.

## 2.2 Prompt Uncertainty

Inspired by uncertainty-based active learning (Siddhant and Lipton, 2018), we aim to select those highly uncertain tasks as the most informative ones at each stage for training. While prior active learning work has proposed numerous uncertainty measurements at the instance level for a single task, these uncertainty values are usually not comparable across tasks. We propose Prompt Uncertainty, a task-level uncertainty measurement that estimates uncertainty values by assessing the disagreement

of the model on the original prediction given complete and perturbed task instructions. By selecting those most prompt-uncertain tasks, we can select the tasks to which the current model is susceptible.

**Prompt Uncertainty Measurement**  Our Prompt Uncertainty method is motivated from *Bayesian Active Learning by Disagreement (BALD)* (Houlsby et al., 2011) in single task Active Learning. Instead of measuring the disagreement among ensemble models in a single task, we measure the disagreement of generation likelihoods on the original prediction over perturbed prompts and original prompts of a task. Figure 2 illustrates the process of measuring the prompt uncertainty of a model to a task's instance $x_0$. To measure the prompt uncertainty $U_t$ for task $t$ given model weights $W$, corresponding unlabeled dataset $X_t$ and instruction (prompt) $I_0^t$, we calculate the average disagreement of likelihood between perturbed and original instruction on $n$ randomly sampled examples from $X_t$.

$$U_t = \frac{1}{n} \sum_{i=1}^{n} \frac{1}{k} \sum_{j=1}^{k} |p_{i,0}^t - p_{i,j}^t|,$$

$$p_{i,j}^t = P(y_i^t | x_i^t, I_j^t, W),$$

$$\text{where } i \in [1, n], j \in [0, k].$$

$P$ is the likelihood of prediction $y$ given model weights $W$, a task instruction $I$ and corresponding task instance $x$. $k$ is the number of perturbations.

For each example $x_i^t \in X^t$, we will first get the original output $y_i^t$ and its corresponding likelihood $p_{i,0}^t$. Then, we will perturb the instruction $k$ times and calculate the average absolute difference between the likelihood of $y_i^t$ given original instruction $p_{i,0}^t$ and perturbed instructions $\{p_{i,j}^t | j \in (1, k)\}$.

In order to perturb a task instruction, it is possible to employ paraphrasing techniques, adding extraneous tokens or randomly omitting words, such that the altered instructions can mostly preserve their meaning (A more detailed discussion can be found in subsection 6.2). In our experiment, we assign a 0.2 drop rate for each word in the instruction to create perturbed instructions. After getting the prompt uncertainty for each remaining task, we will select the highly uncertain ones and add them to the training task pool.

**Underlying Hypothesis**   We describe the underlying hypothesis to propose Prompt Uncertainty. From an uncertainty perspective, when measuring the model's sensitivity toward sampled prompts from a task, we estimate the model's epistemic uncertainty, reflecting the model's lack of knowledge of a particular task. Different from epistemic uncertainty using an ensemble of models (Gal and Ghahramani, 2016), we consider an ensemble of slightly different conditions, i.e., perturbations of prompts for the model, and use the original likelihood to represent the ensembled prediction. From the robustness of the in-context learning perspective, if a model cannot robustly map task instructions to specific latent concepts, which is reflected by the sensitivity regarding perturbations in instructions, its generalization ability to the corresponding task is limited (Xie et al., 2021; Pan et al., 2023). To address this, we hypothesize that training the model on prompt-uncertain tasks will improve its ability to associate prompts with specific latent concepts (tasks), leading to a better zero-shot performance on unseen instructions.

## 3   Experiment Setting

In this work, we experiment with two well-known IT datasets: NIV2 and Self-Instruct (Wang et al., 2022b,a). NIV2 is the largest IT dataset with 1600+ cross-lingual tasks. It focuses on improving model generalization to unseen tasks, while Self-Instruct is used to train the Alpaca model (Taori et al., 2023) and aims to enhance model instruction following ability, following the categorization in prior work (Kung and Peng, 2023). For detailed

| Statistics / IT Models | NIV2 | Self-Instruct |
|---|---|---|
| Dataset | | |
| # of training tasks (instructions) | 756 | 52K |
| # of testing tasks (instructions) | 119 | 252 |
| # of data per task | $\geq 200*$ | 1 |
| Testing on unseen tasks? | ✔ | ✗ |
| Active Instruction Tuning | | |
| # of tasks in initial training set | 68 | 500 |
| # of task to select at $i$th iteration | 68 | $500 * 2^i$ |
| Evaluation | | |
| Evaluation Metrics | Rouge-L | Human Eval GPT Eval |

Table 1: Comparison between NIV2 (Wang et al., 2022b) and Self-Instruct (Wang et al., 2022a) datasets. Most tasks in NIV2 have more than 200 instances, while Self-Instruct only has one instance for each task. These two settings differ in terms of the definition of the task and generalization objective (zero-shot cross-task v.s. cross-task), described in Kung and Peng (2023).

setting and comparison, please see Table 1.

### 3.1   Active Instruction Tuning Setting

**Natural Instruction V2 dataset**   We utilize the NIV2 English tasks split, comprising 756 training tasks and 119 testing tasks, including classification and generative tasks, and run our experiment with **five random seeds**.[2] For each randomized setting, we first randomly select 68 tasks for the initial training set and select another 68 tasks as the validation set, leaving the remaining 620 tasks as the task pool. Afterward, we iteratively apply different task selection strategies to expand the training set and train new IT models, reporting the performance at each iteration [136, 204, 272, 340].

**Self-Instruct dataset**   We first randomly sample 500 tasks as the initial training set from the 52K tasks in the Self-Instruct dataset, leaving the remaining tasks as the remaining task pool. We conduct active instruction tuning and compare model performance at each iteration [1000, 2000, 4000, 8000, 16000].

### 3.2   Task Selection Strategies

Since we are the first to propose active instruction tuning, we construct several baseline task selection strategies: *Random Sampling*, *High Perplexity* and *Low Perplexity*, to compare with our proposed *Prompt Uncertainty* method. *Random Sampling* will randomly sample tasks from the remaining task pool. This is usually a strong baseline in task-selection experiments since we utilize a well-constructed dataset as the task pool, which has less noisy and duplicate data. *High and Low Perplexity* are the baselines inspired by prior active

---

[2]We provide details of our experiments in subsection A.2.

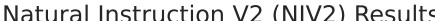

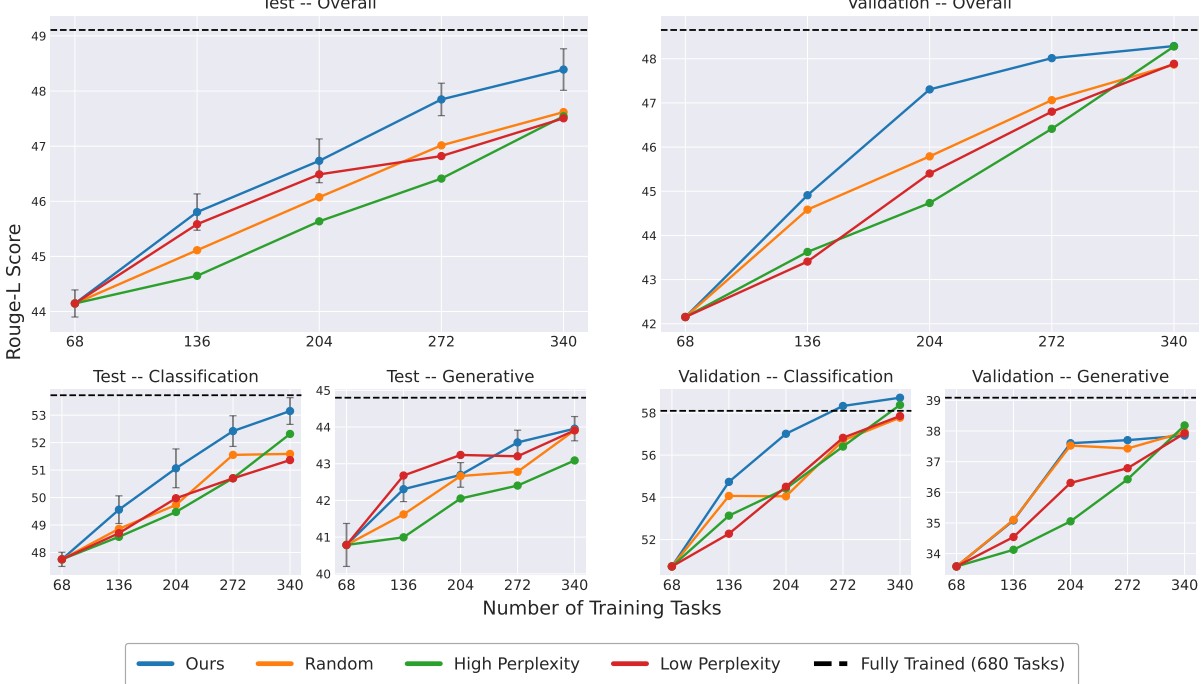

Figure 3: Experiment results for NIV2 dataset. We compare *Ours* prompt uncertainty method with other baselines and report the Rouge-L scores on testing and validation set at each active instruction tuning iteration [136, 204, 272, 340]. **We report the average and standard deviation scores of five runs**, each with a different initial 68 tasks, 68 validation tasks, and 620 remaining task pool. Note that we do not show the standard deviation on validation since each random seed will have a different validation set, leading to high variance. Additionally, we report the *Fully Trained* results which train on the entire task pool. Exact numbers for this experiment can be seen in Table 3 in the Appendix.

learning work, which aims to select difficult/easy tasks by measuring predicted sentence perplexity for generation tasks or entropy for classification tasks. As these uncertainty measurements are established at the instance level, we aggregate the uncertainty score of multiple (ten for NIV2 and one for Self-Instruct) instances in a task to estimate task-level uncertainty. For our method, we measure the *Prompt Uncertainty* using $n = 10$ random examples and $k = 20$ prompt perturbations in NIV2 (refer to section 2). For Self-Instruct, we measure the prompt uncertainty using $n = 1$ random examples and $k = 20$ prompt perturbations.

### 3.3 Training and Evaluation

For NIV2, we follow the current SOTA TK-instruct model's setting, to train the T5-770M model (Raffel et al., 2020) and report the Rouge-L score of *Classification*, *Generative* and *Overall* tasks, on both validation and testing set. During training and testing, we will provide a task definition and two examples as instruction demonstration. For Self-Instruct dataset, we train the LLaMA-7B model (Touvron et al., 2023) follows Alpaca model setting. For evaluation, we report the blind pairwise comparison of each task selection methods

with *Random Sampling* on the 252 user-oriented test set (Wang et al., 2022a). We follow the evaluation in Vicuna (Chiang et al., 2023) to report GPT-4, Chat-GPT (GPT 3.5) and Human evaluation scores, and provide more details in subsection A.1.

## 4 Results

### 4.1 NIV2 Results

Figure 3 displays our experimental results on the NIV2 dataset. For each task selection method, we iteratively select a batch (68) of tasks from the task pool (620 tasks) to train a new model, and compare model performance at each iteration. A better task selection method should achieve consistent superior performance at early iterations, when there are still plenty of tasks to select from. Figure 3 demonstrates that when selecting less than 340 tasks (half of the task pool), our proposed *Prompt Uncertainty* method consistently outperforms other baselines in terms of **Overall** scores for both the validation and testing sets. This shows that training on prompt-uncertain tasks is indeed the most effective way for better zero-shot cross-task generalization ability. On closer examination, our method is highly effective for **Classification** tasks, surpassing all

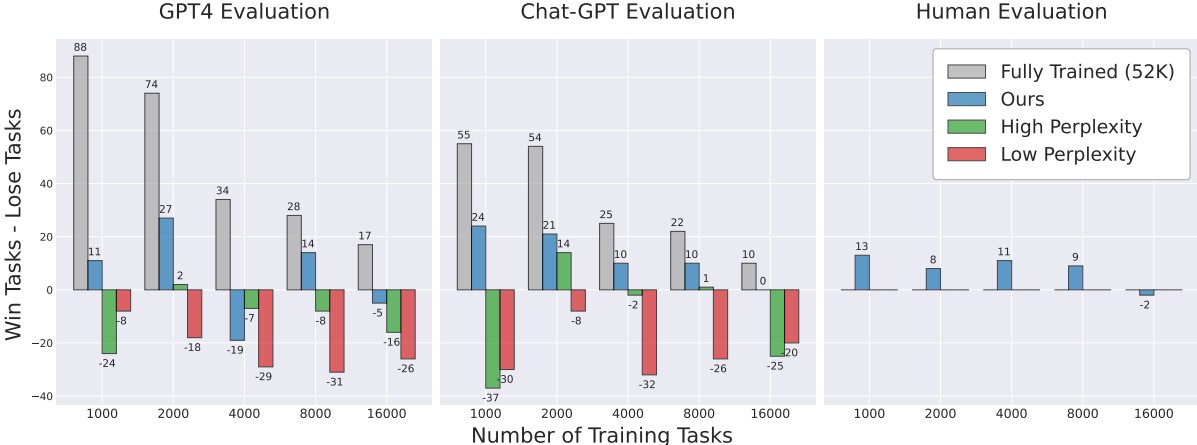

Figure 4: Pairwise comparison results between different task selection strategies and random sampling on 252 user-oriented instruction test set (Wang et al., 2022a), evaluated by GPT4, Chat-GPT, and Human Annotators. For four types of methods at each active instruction tuning iteration: [1000, 2000, 4000, 8000, 16000], we separately conduct a pairwise comparison with *Random Sampling* and report the net winning tasks (Number of Win Tasks - Number of Lose Tasks). Note that we only conduct the human evaluation to compare our proposed *Prompt Uncertainty* to *Random Sampling* method due to high evaluation cost ($600 US Dollars). We provide further details in subsection A.1 Table 4.

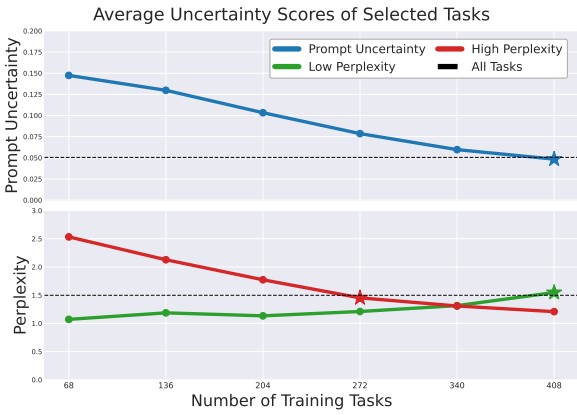

Figure 5: The average uncertainty scores of the selected tasks at each active instruction tuning iteration. For *All Tasks*, we report the average uncertainty score of the 620 training tasks predicted by the initial model.

other baselines. For **Generative** tasks, the *Low Perplexity* method performs well on testing tasks at early iterations but poorly on the validation set. This inconsistency suggests that the model's overall generalizability is not enhanced, but rather the low perplexity tasks in the training set coincidentally benefit the generative tasks in the testing set. Conversely, our proposed method achieves consistently good performance on both testing and validation tasks, outperforming *Random Sampling* on testing tasks and exhibiting similar performance on validation tasks.

We further investigate the trend of uncertainty scores during active instruction tuning. In Figure 5, we illustrate the average uncertainty scores of the selected tasks using different task selection strate-

gies at each iteration. It is shown that when selecting for more than half of the tasks in the training pool, all task selection strategies start deviating and choose tasks with unfavorable uncertainty scores. For example, *High Perplexity* method start selecting tasks with low perplexity scores due to the lack of high perplexity tasks. Specifically, when extending the training tasks from 340 to 408 using *prompt uncertainty*, the average uncertainty score of selected tasks is already slightly lower than that of all tasks at the first iteration, indicating there are no high-uncertainty tasks to select from. Note that the lack of uncertain tasks would occur exclusively in an experimental setting. In practical scenarios where the number of tasks grows rapidly, the exhaustion of uncertain tasks is less likely to happen.

## 4.2 Self-Instruct Results

We show the pairwise preference comparison of all task selection methods against *Random Sampling* in Figure 4. First for *Fully Trained*, we use the official Alpaca release (Taori et al., 2023), which was trained on all 52K tasks. We compare it to *Random Sampling* at each active instruction tuning iteration. It is shown that for both GPT-4 and Chat-GPT evaluation, the *Fully Trained* model outperforms *Random Sampling* with a great margin. However, as more training tasks are randomly sampled, the difference in preferred tasks is diminishing, indicating that IT performance of the Alpaca setting scales with an increasing number of training tasks.

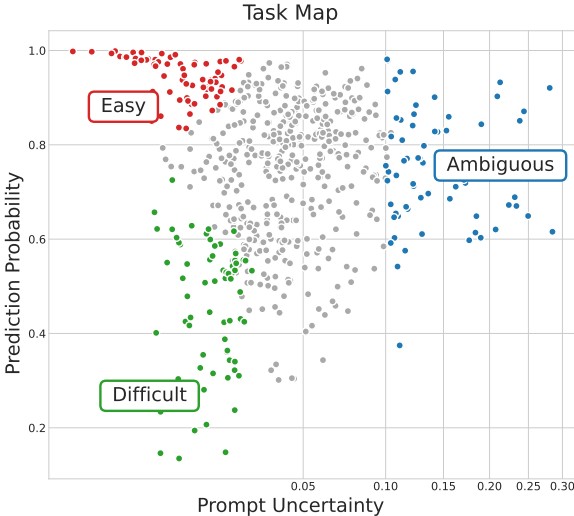

Figure 6: Task Map visualization. We measure the prediction probability and prompt uncertainty of an IT model against 620 tasks in NIV2 and plot the **Ambiguous**, **Easy**, and **Difficult** tasks.

Secondly, for high/low perplexity and our proposed task selection method, we first fine-tune an LLaMA (Touvron et al., 2023) model with 500 tasks and then iteratively extend the number of training tasks to $[1000, 2000, 4000, 8000, 16000]$. We then report the pairwise comparison results against *Random Sampling* at each iteration. Figure 4 shows that *Low Perplexity* and *High Perplexity* are generally subpar with *Random Sampling*, indicating that applying inadequate task selection strategies can hurt the model's performance. In contrast, our *prompt uncertainty* method is almost consistently more preferable by all GPT4, Chat-GPT, and human assessors when selecting less or equal than 8000 tasks, showing that training with prompt-uncertain tasks can lead to better generalization to the user-oriented test tasks. When the number of training tasks increases to 16000, the performance improvement diminishes along with a smaller remaining task pool, which aligns with our results on the NIV2 dataset. Additionally, we discuss our observations regarding applying GPT4, Chat-GPT, and human assessors for pairwise comparisons. It is seen that while the number of net winning tasks (Win task - Lose Tasks) varies a lot across each evaluation method, the overall trend is similar, showing a certain alignment of preference across these automatic or human assessors.

In conclusion, our experiments on NIV2 and Self-Instruct demonstrate that our prompt uncertainty method consistently improves cross-task generalization in two different instruction tuning scenarios, surpassing random sampling and other uncertainty baselines.

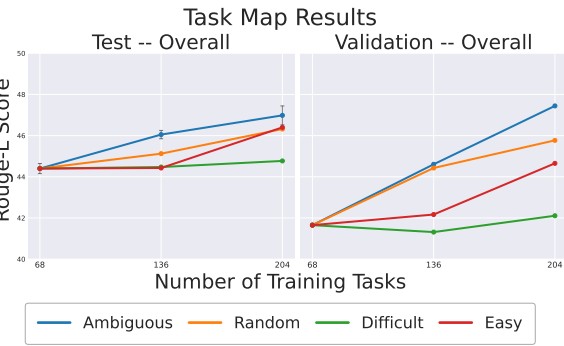

Figure 7: The performance comparison between training on **Ambiguous**, **Easy**, and **Difficult** tasks on the NIV2 dataset. The setting is similar to Figure 3, but we run all methods with three runs for the early active instruction tuning iterations.

## 5 Task Map

Prior work tries to understand a dataset's characteristics in the field of *Dataset Diagnosing*. Motivated by Data Map (Swayamdipta et al., 2020), we propose **Task Map**, a model-based diagnosing tool that understands the contributions of different groups of tasks towards instruction tuning. Different from previous work using data correctness and variability to construct data map, we propose to map tasks with the dimensions of Prediction Probability and Prompt Uncertainty, as in Figure 6. This follows a hypothesis from recent work in explaining in-context learning (ICL) (Xie et al., 2021): when the model performs a task in-context during test time (ICL), it might implicitly map the prompt to a corresponding latent concept and perform the task under the concept. Prediction Probability represents the model's confidence to perform a task, indicating task difficulties. In comparison, Prompt Uncertainty represents the consistency of a model to map a prompt to a certain concept, indicating the task's ambiguity to the model. We further follow the above intuition to categorize the tasks into three types: **Ambiguous** tasks, where models fail to recognize them and have high prompt uncertainty; **Easy** and **Difficult** tasks, where models can map the prompts to a certain latent task knowledge (low prompt uncertainty) and perform the task with high/low confidence (sentence probability), respectively. We then use the tasks from these three categories for instruction tuning on NIV2 (Wang et al., 2022a) to understand the contributions of different groups of tasks.

We show the results in Figure 7. It is seen that while training on **Ambiguous** tasks can effectively improve IT generalization ability and outperform random baseline, training on **Easy** tasks and **Difficult** tasks is generally worse than randomly select-

ing tasks. Furthermore, when selecting more **Easy** tasks can still slightly boost the IT model's performance, **Difficult** tasks can be useless, showing no benefit to the IT model's performance with more training tasks. We hypothesize that **Difficult** tasks can be too specific and hard to learn, therefore useless for improving the IT model's cross-task generalization. While our proposed Task Map can already help diagnose task quality for IT, we look forward to future work conducting a more comprehensive analysis to discuss the role of these task categories to bring a comprehensive understanding of instruction tuning and in-context learning.

## 6 Discussion

### 6.1 Prompt Uncertainty Reflects Task Novelty

To demonstrate how prompt uncertainty reflects the novelty of tasks to a model, we designed a controlled experiment to visualize how the prompt uncertainty scores of tasks change after the model is trained with relevant tasks. To collect a set of relevant tasks, we first gathered eight *Word Analogy* tasks from the NIV2 (Wang et al., 2022b) testing set, which is held unseen from the NIV2 training set. In Figure 8, we measured the prediction probability and prompt uncertainty for 620 unseen tasks (unrelated to analogy tasks) from the NIV2 training set and four of the unseen analogy tasks using an instruction-tuned model, labeled as **M0**, and plotted the task map in blue. We further trained the **M0** model with the other four analogy tasks, resulting in a new model called **M1**, and used it to plot the task map for the 620 irrelevant tasks and four unseen analogy tasks again in orange. It is evident that after training the **M0** model with the four analogy tasks, the overall prompt uncertainty distribution of the 620 irrelevant tasks remains relatively unchanged, while the prompt uncertainty of the four unseen analogy tasks consistently and significantly decreases.[3] This demonstrates that prompt uncertainty can effectively indicate the novelty of tasks within the model. When the model is trained with specific tasks, the prompt uncertainty of those relevant tasks notably decreases. Additionally, please note that the prediction probability does not increase after training with similar tasks for these four analogy tasks. This observation highlights that using prediction probability alone cannot effectively reflect the novelty of tasks.

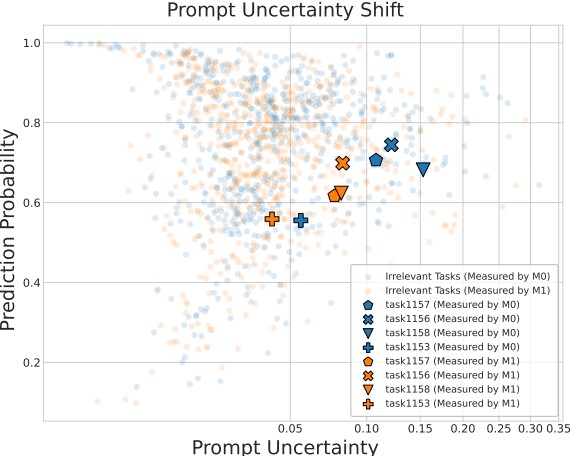

Figure 8: Tasks' prompt uncertainty shifts before and after training with four analogy tasks. We visualize all the tasks on the task map with two models, **M0 (Blue)** and **M1 (Orange)**. **M0** is the same instruction-tuned model as in Figure 6, which does not train on any *analogy tasks*. **M1** is **M0**, further trained with four analogy tasks: *task1159, task1154, task1152, task1155*. Additionally, we measure the prediction probability and prompt uncertainty of 620 irrelevant tasks and four unseen analogy tasks, *task1157, task1156, task1158, task1153*, using both **M0** and **M1**, plotted in orange and blue. It can be seen that after training the model with analogy tasks (from **M0** to **M1**), the prompt uncertainty of the four unseen analogy tasks consistently decreases, while the distribution of other irrelevant tasks remains relatively unchanged.

### 6.2 Prompt Perturbation Methods

While prompt perturbation methods are meant to slightly perturb the prompt without changing its meanings, it is difficult to 100% guarantee the preservation of instruction meaning after automatic paraphrasing methods. To ensure the prompt uncertainty is not measured using an extreme perturbation case, we perturbed all instructions 20 times in our experiments. We also tried several instruction perturbation methods at our early experiment stage, such as randomly repeating tokens or adding extraneous tokens, which achieved similar prompt uncertainty scores as randomly dropping words. Additionally, for the NIV2 and Self-Instruct datasets we used, which have detailed instructions with many redundant tokens (average 56 words per instruction), randomly dropping 20% of tokens will mostly preserve the meaning of the instructions. For other datasets with concise instructions, a higher dropping rate is needed to perturb the instructions, leading to a higher probability of changing instructions meaning entirely.

---

[3]From M0 to M1, the average decrease in prompt uncertainty scores is 0.0018 for the 620 irrelevant tasks and 0.039 for the four analogy tasks. The prompt uncertainty of analogy tasks decreases 21 times more than that of the irrelevant tasks.

## 7 Related Work

### 7.1 Instruction Tuning Paradigm

By training large language models (LLMs) with diverse tasks and corresponding instructions, it allows the model to achieve a decent cross-task generalization ability (Wei et al., 2021; Sanh et al., 2021; Wang et al., 2022b; Taori et al., 2023; Chiang et al., 2023; Ouyang et al., 2022). Following the observation from prior research (Xu et al., 2022; Wang et al., 2022a) that scaling up the number of tasks can significantly improve zero-shot generalization, there is research on continuously adding knowledge to large language models (Scialom et al., 2022; Jang et al., 2023), along with many large-scale IT datasets emerged. Wang et al. (2022a); Wei et al. (2021); Bach et al. (2022); Xu et al. (2022); Jiao et al. (2023) manually augment existing datasets to form large-scale IT datasets and Gupta et al. (2022); Finlayson et al. (2022) manually construct new IT datasets in specific domains. There are also automatic approaches to collecting large-scale IT datasets. Wang et al. (2022a); Honovich et al. (2022) propose generating moderate-quality data from powerful IT models, like GPT-4 and Chat-GPT (OpenAI, 2023). Recently, Dynosaur (Yin et al., 2023a) proposes to curate instructions for the continuously growing huggingface dataset (Lhoest et al., 2021) using GPT-4 to create high-quality IT data with low costs. Additionally, we would like to highlight that as IT models rapidly scale in performance with larger models and datasets, the concern of whether they adhere to instructions still remains (Yin et al., 2023b; Kung and Peng, 2023; Min et al., 2022; Yang et al., 2023; Li et al., 2023; Xue et al., 2023), and requires further investigation. For recent IT development, see (Zhang et al., 2023) for a detailed survey.

### 7.2 Uncertainty Estimation for LLMs

Uncertainty estimation is essential for ensuring safe deployments of neural networks (Abdar et al., 2021). Prior works have decomposed the total uncertainty into aleatoric (data) uncertainty and epistemic (model) uncertainty, and proposed methods to quantify each of them, represented by Monte-Carlo Dropout (Gal and Ghahramani, 2016) and Deep Ensemble (Lakshminarayanan et al., 2017). In particular, data uncertainty measures the intrinsic uncertainty from the data distribution. Model uncertainty measures the uncertainty due to lack of understanding of the task, and can be leveraged to detect adversarial or out-of-distribution data (Feinman et al., 2017; Yin et al., 2022). Recent works have also extended uncertainty quantification to autoregressive language models (Xiao and Wang, 2019; Malinin and Gales, 2020). In this work, we propose a novel epistemic uncertainty measurement for instruction-tuned LLMs by measuring the disagreement of models conditioned on perturbed instructions.

### 7.3 Active Learning and Task Selection

Our work is also related to active learning, which iteratively annotates informative instances from an unlabeled pool for efficient training (Olsson, 2009; Siddhant and Lipton, 2018; Zhang et al., 2022). Strategies for querying informative instances fall into different categories. See Zhang et al. (2022) for a detailed survey. Our method is more related to disagreement-based active learning (Houlsby et al., 2011; Siddhant and Lipton, 2018; Shen et al., 2017), which queries for instances where multiple models disagree the most, and is usually combined with model uncertainty measurements (Gal and Ghahramani, 2016). However, different from active learning which selects informative instances, we consider selections at task-level. We show that simply adopting prior active learning strategies at task-level do not work well and propose our own methods. There are also works doing task selection for specific target tasks (Parvez and Chang, 2021; Zhou et al., 2023). However, we do not assume knowledge of the target task but select tasks solely based on the uncertainty information of the model.

## 8 Conclusion

We propose Active Instruction Tuning with prompt uncertainty, a framework to enhance the generalization ability of the IT model in large-scale instruction tuning. Our experiments on NIV2 and Self-Instruct datasets demonstrate that training on prompt uncertain tasks consistently outperforms random sampling and other uncertainty baselines, highlighting the effectiveness of our approach. We also introduce Task Map, a tool that categorizes tasks based on prompt uncertainty and prediction probability, revealing that while training on ambiguous tasks improves generalization, some difficult tasks offer no benefit. These findings motivate future investigations into prompt uncertainty and task selection strategies for better understanding cross-task generalization and instruction tuning.

## Limitations

While our experiments demonstrate the superiority of our proposed prompt uncertainty method over other baseline task selection methods on the NIV2 and Self-Instruct datasets, there are several limitations to consider. Firstly, our experiments are conducted on open-source instruction tuning models and do not consider the impact of reinforcement learning with human feedback in Instruct-GPT (Ouyang et al., 2022). Secondly, although we conducted our experiments on well-constructed instruction tuning datasets, it is important to note that this setting may not fully capture the challenges posed by noisy or poorly constructed tasks in extreme scenarios, which may require techniques such as noisy task filtering or batch active learning. Lastly, our current experiment on active instruction tuning focuses on comparing task selection methods and does not incorporate the effect of continual learning, which could be valuable for improving IT models in realistic settings. In summary, our work primarily focuses on introducing active instruction tuning and comparing task selection methods within a controlled environment. We look forward to future research to conduct further analysis to comprehensively examine the effects of all these factors.

## Ethics Statement

We describe the computation resources and models we used to conduct our experiments. We conduct all experiments on 4 to 8 48GB NVIDIA A6000 GPUs or 2 to 4 NVIDIA A100 GPUs, along with 48 TB disk storage and AMD EPYC 7413 24-Core Processor. The experiment takes around 5500 GPU hours for one 48GB NVIDIA A6000 GPU. Our experiments do not need to leverage private data. For the model, we use open-sourced Huggingface T5-large-lm-adapt models and LLaMA-7B, Stanford Alpaca-7B for our experiments, and we will release our code once the paper is accepted.

## Acknowledgements

We would like to thank Hritik Bansal and Da Yin for their valuable insights during discussion, paper reviews, and constructive comments. We thank the anonymous reviewers for their feedback. This work was partially supported by AFOSR MURI via Grant #FA9550-22-1-0380, Defense Advanced Research Project Agency (DARPA) grant #HR00112290103/HR0011260656, CISCO and ONR grant #N00014-23-1-2780.

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

# A Appendix

## A.1 Evaluation details

**Human Evaluation** For human evaluation in section 4 and Figure 4, we recruit crowd-source workers on Amazon Mechanical Turk who are native English speakers, score at least 70% on a qualification test, and pass the attention test. For the annotation task, three annotators are presented with the task instruction, the input, and the expected output, followed by two models' outputs in random order. The annotators are asked to indicate whether the first model wins, loses, or has a tie. An example of the annotation interface is presented Figure 9. The final comparison decisions are aggregated from the raw annotations using majority voting. We assign a tie label when all the annotators disagree. To calculate the inter-annotator agreement, we define no-contradiction as agreement with the tie entries removed since an annotator slightly supporting a model may also vote for a tie. Under this definition, the no-contradiction rate is measured as 60.4%. Among the cases with contradiction, we find two annotators agree for 82% cases, and all annotators disagree in only 18% cases. We set the per-item reward to $0.1 to reach an hourly rate of $15. We collect 3780 comparison annotations to compare Prompt Uncertainty with *Random Sampling* at each active instruction tuning iteration. The total annotation cost is approximately 600 US Dollars.

**GPT-4/Chat-GPT Evaluation** We conduct a blind pairwise comparison on GPT-4 and Chat-GPT (GPT-3.5) models using Open-AI API, following a similar template as we used for human evaluation, which is shown in Table 2. To compare two models on one instance, we will randomly assign "(1)" and "(2)" to the model's predictions and prompt the model to reply with the better choice or "Equal" if two predictions are equally good. Note that when applying GPT evaluation, there are very rare cases (about 0.7% of the cases) that the GPT models will reply with unrelated output, which we will assign "Equal" to these instances. The total annotation cost is approximately 50 US Dollars for GPT-4 and 2 US Dollars for Chat-GPT. For full evaluation results, please refer to Table 4

## A.2 Experiment details

We provide the details of our experiments on two well-known instruction tuning datasets: NIV2 (Wang et al., 2022b) and Self-Instruct dataset (Wang et al., 2022a).

**NIV2 - Active Instruction Tuning Details** We utilize the NIV2 English tasks split, comprising 756 training tasks and 119 testing tasks, including classification and generative tasks. We employ five random seeds without selection in our active instruction tuning experiment. Each seed involves randomly sampling 68 tasks as initial training tasks and 68 tasks as validation tasks. The remaining 620 training tasks form the remaining task pool. In each active learning iteration, we maintain a fixed classification and generative task ratio and select 24 classification tasks and 44 generative tasks using different task selection strategies. This fixed ratio allows a more meaningful comparison of our results as we evaluate overall, classification, and generative task scores separately. After the new tasks are sampled, we add them to the previously selected training tasks and form a new training task set. We further train a new instruction tuning model with the updated training task set.

**Self-Instruct - Active Instruction Tuning Details** We utilize the 52K self-instruct dataset as the task pool. For the active instruction tuning experiment, we will randomly sample 500 tasks as the initial training set and further compare model performance at $[1000, 2000, 4000, 8000, 16000]$ training tasks. For task selection, we will first divide all tasks into 13 chunks by output sequence length $[[1, 10], [11, 20], ..., [121, 130]]$, and then apply the task selection methods on each chunk of tasks, following the ratio of the number of tasks in all chunks. We conduct this extra step to normalize the output sequence length of the selected task for each task selection method. This ensures there is no imbalance in output sequence length during task selection.

**Training Details** For experiments on NIV2 dataset (Wang et al., 2022b), we follow the TK-instruct setting, the SOTA model on the NIV2 dataset to train the T5-770M model (Raffel et al., 2020) with learning rate 2e-5, batch size 128 and 200 instances per task for eight epochs. We evaluate the model's zero-shot performance on the validation set at each epoch and select the model checkpoint with the best validation score. For evaluation, we follow (Kung and Peng, 2023) setting to report the Rouge-L score of *Overall*, *Classification*, and *Generative* tasks on both validation and testing sets. For experiments on Self-Instruct dataset (Wang et al., 2022a), We follow Alpaca's

settings to train the LLaMA-7B model with learning rate 2e-5, batch size 128 for four epochs.

**Computing Resources** For the experiment on NIV2 dataset (Wang et al., 2022b), we conduct our experiments using 4 to 8 Nvidia 48GB A6000 GPUs. For each uncertainty method, it takes around 1200 GPU hours, a total of 5000 GPU hours(for a single GPU), to run all experiments for Figure 3. For the experiment on Self-Instruct dataset (Wang et al., 2022a), we run with 2 Nvidia 80GB A100 GPUs. Each uncertainty method takes around 40 GPU hours, which sums to 160 GPU hours for all experiments in Figure 4.

*Given the model predictions from Random Sampling and Prompt Uncertainty which trained with 1000 tasks, we first randomly assign "(1)" and "(2)" tags:*
**(1)**: Prompt-Uncertainty-1000
**(2)**: Random-Sampling-1000
*Now we fill the GPT prompt template with these two model predictions.*

**GPT Prompt:**
You're a helpful AI system that is meant to imitate human judgment.
Given an *instruction*, *input*, and two *predictions* "(1)" and "(2)", please tell me which prediction is most satisfying and correct.
If (1) is better, output "(1)".
If (2) is better, output "(2)".
If both predictions are equally good or equally bad, you can output "Equal".

**Instruction:**
If you could help me write an email to my friends inviting them to dinner on Friday, it would be greatly appreciated.

**Input:**

**Now given two predictions:**

**(1)**: Dear Friends,
I would like to invite you all to join me for dinner on Friday. I am planning to make some of my favorite dishes, and I would love for you all to join me.

The dinner will be held at my place at 7pm on Friday, and I am looking forward to seeing you all there.

Please let me know if you are able to make it so that I can make sure to have enough food.

See you then!

Sincerely,

[Your Name]

**(2)**: Dear [Name],

With sincerity,
[Your Name]

**Output:**

**GPT Output:** (1)

Table 2: We show our GPT evaluation template. When comparing two models' predictions, we will randomly assign "(1)" or "(2)" tags and then fill them into the GPT templates with provided *instruction* and *inputs*. In this specific instance, the input is an empty string. All the underlined text are the component we injected into the template.

Figure 9: An example of the annotation interface for the human evaluation in §4.2.

**NIV2 Results (Rouge-L) – Test Set**

| Task Selection Methods | Task Num | Overall | | | | | | | Classification | | | | | | | Generative | | | | | | |
|---|---|---|---|---|---|---|---|---|---|---|---|---|---|---|---|---|---|---|---|---|---|---|
| Random Seeds → | | 10 | 20 | 30 | 40 | 60 | Avg | Std | 10 | 20 | 30 | 40 | 60 | Avg | Std | 10 | 20 | 30 | 40 | 60 | Avg | Std |
| Fully Trained | 680 | 50.51 | 48.96 | 49.41 | 47.27 | 49.39 | 49.11 | 1.18 | 57.03 | 53.35 | 53.99 | 50.26 | 54.00 | 53.73 | 2.41 | 44.43 | 44.87 | 45.13 | 44.47 | 45.09 | 44.80 | 0.33 |
| Prompt Uncertainty | 68 | 44.67 | 43.71 | 44.82 | 43.78 | 43.75 | 44.15 | 0.55 | 46.80 | 47.69 | 48.25 | 47.84 | 48.16 | 47.75 | 0.58 | 42.68 | 40.00 | 41.62 | 40.00 | 39.63 | 40.79 | 1.31 |
| | 136 | 45.55 | 46.54 | 46.06 | 44.65 | 46.22 | 45.80 | 0.74 | 48.34 | 50.83 | 49.59 | 48.52 | 50.51 | 49.56 | 1.13 | 42.95 | 42.54 | 42.77 | 41.04 | 42.23 | 42.31 | 0.76 |
| | 204 | 46.44 | 48.28 | 46.23 | 46.09 | 46.63 | 46.73 | 0.89 | 50.95 | 53.74 | 49.80 | 50.85 | 49.98 | 51.06 | 1.58 | 42.23 | 43.20 | 42.89 | 41.65 | 43.50 | 42.69 | 0.75 |
| | 272 | 47.38 | 48.64 | 47.25 | 47.49 | 48.48 | 47.85 | 0.66 | 51.92 | 54.05 | 50.65 | 52.73 | 52.75 | 52.42 | 1.25 | 43.14 | 43.58 | 44.08 | 42.61 | 44.49 | 43.58 | 0.74 |
| | 340 | 48.12 | 48.93 | 47.23 | 49.44 | 48.23 | 48.39 | 0.84 | 52.93 | 54.08 | 51.92 | 54.45 | 52.37 | 53.15 | 1.09 | 43.63 | 44.13 | 42.86 | 44.78 | 44.37 | 43.95 | 0.74 |
| Random Sampling | 68 | 44.67 | 43.71 | 44.82 | 43.78 | 43.75 | 44.15 | 0.55 | 46.80 | 47.69 | 48.25 | 47.84 | 48.16 | 47.75 | 0.58 | 42.68 | 40.00 | 41.62 | 40.00 | 39.63 | 40.79 | 1.31 |
| | 136 | 45.07 | 45.75 | 44.55 | 45.04 | 45.15 | 45.11 | 0.43 | 48.45 | 50.50 | 47.82 | 47.97 | 49.56 | 48.86 | 1.14 | 41.92 | 41.32 | 41.50 | 42.31 | 41.04 | 41.62 | 0.50 |
| | 204 | 46.15 | 46.87 | 45.94 | 46.15 | 45.26 | 46.07 | 0.58 | 49.76 | 51.36 | 49.08 | 48.73 | 49.72 | 49.73 | 1.01 | 42.79 | 42.68 | 43.02 | 43.74 | 41.09 | 42.66 | 0.97 |
| | 272 | 47.90 | 48.24 | 46.44 | 46.24 | 46.25 | 47.01 | 0.97 | 53.90 | 53.38 | 50.34 | 50.00 | 50.15 | 51.55 | 1.92 | 42.30 | 43.44 | 42.81 | 42.74 | 42.61 | 42.78 | 0.42 |
| | 340 | 47.37 | 48.35 | 47.82 | 47.22 | 47.33 | 47.62 | 0.47 | 50.12 | 52.74 | 52.61 | 50.54 | 51.94 | 51.59 | 1.20 | 44.81 | 44.26 | 43.35 | 44.11 | 43.02 | 43.91 | 0.72 |
| High Perplexity | 68 | 44.67 | 43.71 | 44.82 | 43.78 | 43.75 | 44.15 | 0.55 | 46.80 | 47.69 | 48.25 | 47.84 | 48.16 | 47.75 | 0.58 | 42.68 | 40.00 | 41.62 | 40.00 | 39.63 | 40.79 | 1.31 |
| | 136 | 45.15 | 43.97 | 45.31 | 43.98 | 44.83 | 44.65 | 0.64 | 48.41 | 49.04 | 48.87 | 48.56 | 47.96 | 48.57 | 0.42 | 42.10 | 39.25 | 42.00 | 39.71 | 41.91 | 40.99 | 1.39 |
| | 204 | 45.50 | 46.91 | 45.43 | 44.19 | 46.15 | 45.64 | 1.00 | 49.13 | 51.29 | 49.25 | 47.89 | 49.81 | 49.47 | 1.23 | 42.11 | 42.82 | 41.86 | 40.73 | 42.74 | 42.05 | 0.84 |
| | 272 | 47.57 | 46.09 | 47.33 | 44.80 | 46.27 | 46.41 | 1.11 | 53.19 | 50.10 | 52.29 | 48.73 | 49.21 | 50.70 | 1.95 | 42.32 | 42.35 | 42.70 | 41.12 | 43.53 | 42.40 | 0.87 |
| | 340 | 48.65 | 48.59 | 48.15 | 45.66 | 46.66 | 47.54 | 1.32 | 53.96 | 53.71 | 54.57 | 49.19 | 50.15 | 52.32 | 2.46 | 43.70 | 43.81 | 42.16 | 42.36 | 43.41 | 43.09 | 0.77 |
| Low Perplexity | 68 | 44.67 | 43.71 | 44.82 | 43.78 | 43.75 | 44.15 | 0.55 | 46.80 | 47.69 | 48.25 | 47.84 | 48.16 | 47.75 | 0.58 | 42.68 | 40.00 | 41.62 | 40.00 | 39.63 | 40.79 | 1.31 |
| | 136 | 45.15 | 45.99 | 46.13 | 45.55 | 45.10 | 45.58 | 0.47 | 46.99 | 48.82 | 48.72 | 49.10 | 49.88 | 48.70 | 1.06 | 43.43 | 43.36 | 43.72 | 42.24 | 40.64 | 42.68 | 1.27 |
| | 204 | 46.21 | 46.21 | 46.22 | 47.79 | 46.01 | 46.49 | 0.73 | 49.63 | 48.83 | 48.83 | 51.63 | 50.95 | 49.97 | 1.27 | 43.02 | 43.76 | 43.79 | 44.21 | 41.41 | 43.24 | 1.11 |
| | 272 | 46.71 | 45.93 | 47.43 | 47.32 | 46.71 | 46.82 | 0.60 | 50.77 | 48.57 | 51.87 | 51.39 | 50.89 | 50.70 | 1.27 | 42.93 | 43.48 | 43.29 | 43.51 | 42.80 | 43.20 | 0.32 |
| | 340 | 47.86 | 48.34 | 47.29 | 47.57 | 46.47 | 47.51 | 0.70 | 52.64 | 51.46 | 50.66 | 51.92 | 50.14 | 51.36 | 0.99 | 43.40 | 45.43 | 44.15 | 43.51 | 43.05 | 43.91 | 0.94 |

**NIV2 Results (Rouge-L) – Validation Set**

| Task Selection Methods | Task Num | Overall | | | | | | | Classification | | | | | | | Generative | | | | | | |
|---|---|---|---|---|---|---|---|---|---|---|---|---|---|---|---|---|---|---|---|---|---|---|
| Random Seeds → | | 10 | 20 | 30 | 40 | 60 | Avg | Std | 10 | 20 | 30 | 40 | 60 | Avg | Std | 10 | 20 | 30 | 40 | 60 | Avg | Std |
| Full | 680 | 50.86 | 48.24 | 48.28 | 48.36 | 47.52 | 48.65 | 1.28 | 58.66 | 59.76 | 54.86 | 57.52 | 59.68 | 58.10 | 2.02 | 43.04 | 36.47 | 41.71 | 38.94 | 35.27 | 39.09 | 3.31 |
| Prompt Uncertainty | 68 | 42.27 | 41.28 | 41.40 | 44.66 | 41.15 | 42.15 | 1.47 | 44.42 | 53.67 | 46.83 | 53.32 | 55.39 | 50.73 | 4.80 | 40.11 | 28.90 | 35.97 | 36.01 | 26.92 | 33.58 | 5.49 |
| | 136 | 44.78 | 43.35 | 45.69 | 48.16 | 42.57 | 44.91 | 2.19 | 49.29 | 57.21 | 52.76 | 57.93 | 56.47 | 54.73 | 3.64 | 40.27 | 29.48 | 38.61 | 38.38 | 28.67 | 35.08 | 5.54 |
| | 204 | 47.75 | 47.60 | 46.98 | 50.03 | 44.17 | 47.31 | 2.10 | 53.69 | 60.83 | 53.21 | 60.31 | 57.01 | 57.01 | 3.57 | 41.81 | 34.36 | 40.75 | 39.75 | 31.34 | 37.60 | 4.53 |
| | 272 | 47.69 | 47.98 | 48.13 | 49.73 | 46.54 | 48.01 | 1.14 | 55.5 | 61.03 | 54.45 | 60.80 | 59.85 | 58.33 | 3.11 | 39.88 | 34.93 | 41.81 | 38.66 | 33.23 | 37.70 | 3.54 |
| | 340 | 48.12 | 48.00 | 48.39 | 49.61 | 47.31 | 48.29 | 0.84 | 56.78 | 60.31 | 54.92 | 60.79 | 60.79 | 58.72 | 2.71 | 39.45 | 35.70 | 41.85 | 38.43 | 33.82 | 37.85 | 3.15 |
| Random Sampling | 68 | 42.27 | 41.28 | 41.40 | 44.66 | 41.15 | 42.15 | 1.47 | 44.42 | 53.67 | 46.83 | 53.32 | 55.39 | 50.73 | 4.80 | 40.11 | 28.90 | 35.97 | 36.01 | 26.92 | 33.58 | 5.49 |
| | 136 | 45.31 | 44.36 | 43.61 | 47.39 | 42.25 | 44.58 | 1.93 | 50.75 | 56.15 | 50.21 | 56.77 | 56.45 | 54.07 | 3.29 | 39.87 | 32.57 | 37.01 | 38.01 | 28.06 | 35.10 | 4.77 |
| | 204 | 45.99 | 45.54 | 45.78 | 47.73 | 43.91 | 45.79 | 1.36 | 49.09 | 56.43 | 51.56 | 56.74 | 56.41 | 54.05 | 3.51 | 42.88 | 34.64 | 39.99 | 38.72 | 31.40 | 37.53 | 4.53 |
| | 272 | 47.38 | 47.13 | 45.99 | 48.76 | 46.05 | 47.06 | 1.14 | 52.79 | 60.45 | 52.58 | 58.61 | 59.03 | 56.69 | 3.72 | 41.98 | 33.80 | 39.41 | 38.90 | 33.06 | 37.43 | 3.84 |
| | 340 | 47.98 | 48.89 | 47.80 | 49.52 | 45.15 | 47.87 | 1.67 | 54.80 | 61.90 | 54.71 | 60.50 | 56.95 | 57.77 | 3.29 | 41.17 | 35.88 | 40.89 | 38.54 | 33.34 | 37.96 | 3.35 |
| High Perplexity | 68 | 42.27 | 41.28 | 41.40 | 44.66 | 41.15 | 42.15 | 1.47 | 44.42 | 53.67 | 46.83 | 53.32 | 55.39 | 50.73 | 4.80 | 40.11 | 28.90 | 35.97 | 36.01 | 26.92 | 33.58 | 5.49 |
| | 136 | 43.43 | 43.79 | 41.51 | 47.04 | 42.36 | 43.63 | 2.11 | 48.34 | 55.35 | 50.13 | 56.09 | 55.75 | 53.13 | 3.62 | 38.51 | 32.24 | 32.90 | 38.00 | 28.98 | 34.13 | 4.05 |
| | 204 | 44.11 | 46.97 | 41.89 | 47.07 | 43.63 | 44.73 | 2.24 | 48.75 | 58.62 | 50.16 | 56.65 | 57.89 | 54.41 | 4.61 | 39.47 | 35.32 | 33.62 | 37.49 | 29.37 | 35.05 | 3.87 |
| | 272 | 49.59 | 44.64 | 44.29 | 49.02 | 44.53 | 46.41 | 2.65 | 56.83 | 55.59 | 52.85 | 58.34 | 58.41 | 56.40 | 2.30 | 42.35 | 33.70 | 35.73 | 39.69 | 30.65 | 36.42 | 4.66 |
| | 340 | 50.14 | 48.74 | 47.37 | 50.35 | 44.79 | 48.28 | 2.29 | 56.30 | 62.06 | 56.41 | 60.38 | 56.73 | 58.38 | 2.67 | 43.98 | 35.43 | 38.33 | 40.31 | 32.86 | 38.18 | 4.30 |
| Low Perplexity | 68 | 42.27 | 41.28 | 41.40 | 44.66 | 41.15 | 42.15 | 1.47 | 44.42 | 53.67 | 46.83 | 53.32 | 55.39 | 50.73 | 4.80 | 40.11 | 28.90 | 35.97 | 36.01 | 26.92 | 33.58 | 5.49 |
| | 136 | 43.85 | 43.80 | 41.21 | 46.77 | 41.41 | 43.41 | 2.26 | 47.87 | 55.52 | 47.05 | 56.04 | 54.90 | 52.28 | 4.42 | 39.83 | 32.09 | 35.38 | 37.49 | 27.92 | 34.54 | 4.67 |
| | 204 | 46.62 | 44.17 | 42.88 | 49.49 | 43.85 | 45.40 | 2.67 | 50.72 | 56.68 | 49.53 | 59.04 | 56.53 | 54.50 | 4.14 | 42.53 | 31.66 | 36.23 | 39.94 | 31.18 | 36.31 | 5.00 |
| | 272 | 47.22 | 46.02 | 46.83 | 49.64 | 44.30 | 46.80 | 1.94 | 53.95 | 57.95 | 54.37 | 60.29 | 57.53 | 56.82 | 2.65 | 40.49 | 34.08 | 39.30 | 39.00 | 31.06 | 36.79 | 4.03 |
| | 340 | 47.08 | 48.85 | 47.58 | 50.47 | 45.45 | 47.89 | 1.89 | 55.12 | 61.16 | 53.75 | 61.23 | 57.85 | 57.99 | 3.42 | 39.04 | 36.54 | 41.42 | 39.71 | 32.91 | 37.92 | 3.31 |

Table 3: Full experiment results in Figure 3.

| GPT-4 Evaluation (Compare to *Random Sampling*) | | | | | | |
|---|---|---|---|---|---|---|
| Task Selection Methods | Task Num | Win | Lose | Tie | Error | **Win - Lose*** |
| *Fully Trained* | 1000 | 143 | 55 | 54 | 0 | 88 |
| | 2000 | 129 | 55 | 67 | 1 | 74 |
| | 4000 | 104 | 70 | 78 | 0 | 34 |
| | 8000 | 95 | 67 | 90 | 0 | 28 |
| | 16000 | 84 | 67 | 101 | 0 | 17 |
| *Prompt Uncertainty (Ours)* | 1000 | 83 | 72 | 97 | 0 | 11 |
| | 2000 | 100 | 73 | 79 | 0 | 27 |
| | 4000 | 75 | 94 | 83 | 0 | 19 |
| | 8000 | 89 | 75 | 88 | 0 | 14 |
| | 16000 | 78 | 83 | 91 | 0 | -5 |
| *High Perplexity* | 1000 | 64 | 88 | 100 | 0 | 24 |
| | 2000 | 83 | 81 | 88 | 0 | 2 |
| | 4000 | 77 | 84 | 91 | 0 | -7 |
| | 8000 | 80 | 88 | 84 | 0 | -8 |
| | 16000 | 70 | 86 | 96 | 0 | 16 |
| *Low Perplexity* | 1000 | 73 | 81 | 98 | 0 | -8 |
| | 2000 | 78 | 96 | 78 | 0 | 18 |
| | 4000 | 74 | 03 | 75 | 0 | 29 |
| | 8000 | 73 | 04 | 75 | 0 | 31 |
| | 16000 | 71 | 97 | 84 | 0 | 26 |
| Chat-GPT Evaluation (Compare to *Random Sampling*) | | | | | | |
| Task Selection Methods | Task Num | Win | Lose | Tie | Error | **Win - Lose*** |
| *Fully Trained* | 1000 | 117 | 62 | 71 | 2 | 55 |
| | 2000 | 125 | 71 | 55 | 1 | 54 |
| | 4000 | 99 | 74 | 76 | 3 | 25 |
| | 8000 | 93 | 71 | 86 | 2 | 22 |
| | 16000 | 82 | 72 | 98 | 0 | 10 |
| *Prompt Uncertainty (Ours)* | 1000 | 94 | 70 | 88 | 0 | 24 |
| | 2000 | 94 | 73 | 82 | 3 | 21 |
| | 4000 | 86 | 76 | 90 | 0 | 10 |
| | 8000 | 86 | 76 | 88 | 2 | 10 |
| | 16000 | 83 | 83 | 86 | 0 | 0 |
| *High Perplexity* | 1000 | 58 | 95 | 98 | 1 | -37 |
| | 2000 | 88 | 74 | 90 | 0 | 14 |
| | 4000 | 84 | 86 | 82 | 0 | -2 |
| | 8000 | 82 | 81 | 84 | 5 | 1 |
| | 16000 | 68 | 93 | 91 | 0 | -25 |
| *Low Perplexity* | 1000 | 68 | 98 | 86 | 0 | -30 |
| | 2000 | 82 | 90 | 79 | 1 | -8 |
| | 4000 | 71 | 103 | 74 | 4 | -32 |
| | 8000 | 80 | 106 | 66 | 0 | -26 |
| | 16000 | 76 | 96 | 79 | 1 | -20 |
| Human Evaluation (Compare to *Random Sampling*) | | | | | | |
| Task Selection Methods | Task Num | Win | Lose | Tie | Error | **Win - Lose*** |
| *Prompt Uncertainty (Ours)* | 1000 | 101 | 88 | 13 | – | 13 |
| | 2000 | 74 | 66 | 8 | – | 8 |
| | 4000 | 94 | 83 | 11 | – | 11 |
| | 8000 | 93 | 84 | 9 | – | 9 |
| | 16000 | 91 | 93 | -2 | – | -2 |

Table 4: Full experiment results in Figure 4.