# OpenReview forum: "Active Instruction Tuning: Improving Cross-Task Generalization by Training on Prompt Sensitive Tasks"
_EMNLP/2023/Conference — EMNLP 2023 Main_

### Official Review · Reviewer_Yx6p · 2023-08-01

**Soundness:** 4

**Excitement:**

3: Ambivalent: It has merits (e.g., it reports state-of-the-art results, the idea is nice), but there are key weaknesses (e.g., it describes incremental work), and it can significantly benefit from another round of revision. However, I won't object to accepting it if my co-reviewers champion it.

**Paper Topic And Main Contributions:**

This paper proposes a prompt-uncertainty-based task selection strategy that actively selects tasks that necessitate instruction tuning to mitigate the issue of extensive training resource cost caused by the increasing amount of data. Experimental results show that the proposed strategy can lead to better generalization over NIV2 and Self-instruct benchmark. The task map further demonstrates the effectiveness of selecting tasks with prompt uncertainty.

**Questions For The Authors:**

The data size of the selected task can also impact the training process, as the iteration steps for updating parameters may vary significantly. Is this a factor that needs to be considered?

**Reasons To Accept:**

1. It's novel to select tasks with prompt uncertainty and experimental results show its superiority compared to the perplexity-based method.
2. Task map that combines prediction probability and prompt uncertainty can be useful to select Ambiguous tasks to perform instruction tuning and yield a better generalization.

**Reasons To Reject:**

There is still a certain gap between the overall performance of the proposed schema and fully trained instruction tuning. While the purpose of instruction tuning is to obtain better generalization, although training on only about half of the tasks can achieve comparable results with the method, I think most people will still choose to train on all data.

**Reproducibility:**

3: Could reproduce the results with some difficulty. The settings of parameters are underspecified or subjectively determined; the training/evaluation data are not widely available.

**Reviewer Confidence:**

4: Quite sure. I tried to check the important points carefully. It's unlikely, though conceivable, that I missed something that should affect my ratings.

---

> ### Author Rebuttal · Authors · 2023-08-29
>
> We thank the reviewer for the constructive feedback.
>
> **Performance gap between proposed schema and fully trained instruction tuning**:
> While surpassing a fully trained instruction tuning model is the ultimate goal, being able to select the most effective tasks to train on a given limited computation budget has many practical applications. In a real-world context, with vast tasks and data, while large corporations may be able to train on all existing tasks, small corporations may find training on all tasks impractical.
> Similar to prior active learning work aimed to improve annotation/computing efficiency, our proposed active instruction tuning framework aims to make instruction tuning more applicable by finding efficient task selection methods and bringing further understanding to task characteristics.
>
> **Question: Is each task's data size a factor that needs to be considered?**:
> Yes. We believe each task's data size can be a factor for instruction tuning performance on the NIV2 dataset. Since all tasks can have different difficulties and require different numbers of training instances, selecting the best amount of instances for each task can potentially improve instruction tuning efficiency and achieve better overall generalizability. However, since there are no clear task boundaries for the Self-Instruct dataset, each task/instruction only has one instance, so this is not considered a factor.
> In our work, we fix the number of instances (200) used in each task for NIV2 to focus on task selection. Additionally, there are some potential methods for future research on determining the number of instances used in each task. For example, if we select tasks from the task pool without replacement (put them back into the task pool after selection) with only a few instances for each selection iteration, we can eventually select different amounts of instances from each task. We also look forward to future work research on the appropriate amount of instances for each task during instruction tuning, which can potentially improve active instruction tuning further.

---

### Official Review · Reviewer_wiNR · 2023-08-04

**Soundness:** 3

**Excitement:**

4: Strong: This paper deepens the understanding of some phenomenon or lowers the barriers to an existing research direction.

**Missing References:**

see reasons for reject

**Paper Topic And Main Contributions:**

The paper proposes a framework called Active Instruction Tuning for improving the cross-task generalization of large language models (LLMs) trained on diverse tasks with instructions. The key idea is to actively select informative tasks based on prompt uncertainty, which measures the sensitivity of the model against perturbed prompts. The authors conduct experiments on two instruction tuning datasets and demonstrate that their method consistently outperforms baseline strategies for task selection, achieving better generalization with fewer training tasks. They also introduce a task map that categorizes tasks based on prompt uncertainty and prediction probability, providing insights into task characteristics and quality.

**Questions For The Authors:**

see reasons for reject

**Reasons To Accept:**

1) The paper proposes a novel framework for active instruction tuning, which addresses the challenge of task selection to improve cross-task generalization in LLMs.
2) The prompt uncertainty measurement is a unique contribution that allows for the selection of informative tasks based on the model's sensitivity to prompt perturbations.
3) The experiments are well-designed and conducted on two different datasets, demonstrating the effectiveness of the proposed method.

**Reasons To Reject:**

1) The training process appears to involve the use of an updated task pool. It would be of interest to know whether the model is retrained by shuffling this updated task pool, or whether it is trained in a continual learning manner, focusing only on the newly added task. If it is the former, I suspect this could be a time-consuming process.
2) The authors exclusively compared their method with their own established baseline, without including well-known active learning methods from previous studies. Despite the authors' assertion that their method operates on a task level, it remains unclear why other baseline methods could not be applied in the same context.
3) The authors omitted a reference to the NIV2 dataset in the Introduction section, which is a noticeable absence.

**Reproducibility:**

4: Could mostly reproduce the results, but there may be some variation because of sample variance or minor variations in their interpretation of the protocol or method.

**Reviewer Confidence:**

3: Pretty sure, but there's a chance I missed something. Although I have a good feel for this area in general, I did not carefully check the paper's details, e.g., the math, experimental design, or novelty.

**Typos Grammar Style And Presentation Improvements:**

Overall, the paper offers a comprehensive overview of the active instruction tuning framework and highlights intriguing insights regarding the efficacy of prompt uncertainty for task selection. However, the distinction between task-level and instance-level selection remains somewhat unclear in the paper. I am left wondering whether it would be more straightforward to consider the selection process on an instance-level, given that each instance is encompassed by its own instruction.

---

> ### Author Rebuttal · Authors · 2023-08-29
>
> We thank the reviewer for the constructive feedback.
>
> **Whether the model is retrained by shuffling the updated task pool or in a continual learning manner**:
> We follow a similar experiment setting as prior active learning work in a single-task scenario to re-train a new model for each active instruction tuning iteration.
> We do not incorporate continual learning to investigate historical task performance but focus only on generalization performance, which better aligns with the goal of instruction tuning.
> In terms of efficiency, we acknowledge the limitation of this strategy in our limitation section (lines 585 - 588).
> We leave the combination of continual learning and active instruction tuning as an exciting future work.
>
> **No well-known active learning methods as baselines**:
> Previous Active Learning (AL) methods primarily work on multi-class classification tasks, while IT is generative, which obscures most methods to be directly applied.
> To establish baselines for active instruction tuning, we apply perplexity, which is one of the previous AL methods for instance-level selection. Aggregating perplexity to the task level is a straightforward extension of instance-level AL to task-level AL. We show that our prompt uncertainty is better than this baseline.
> We will add more detailed descriptions in our paper for better clarity.
>
> **Omitted reference to the NIV2 dataset in introduction**:
> Thanks for noticing! We will add a reference in the updated version.

---

### Official Review · Reviewer_26oa · 2023-08-13

**Soundness:** 3

**Excitement:**

4: Strong: This paper deepens the understanding of some phenomenon or lowers the barriers to an existing research direction.

**Missing References:**

- Scialom, T., Chakrabarty, T., & Muresan, S. (2022, December). Fine-tuned language models are continual learners. In Proceedings of the 2022 Conference on Empirical Methods in Natural Language Processing (pp. 6107-6122).

- Jang, J., Kim, S., Ye, S., Kim, D., Logeswaran, L., Lee, M., ... & Seo, M. (2023). Exploring the benefits of training expert language models over instruction tuning. arXiv preprint arXiv:2302.03202.



**Paper Topic And Main Contributions:**

The authors introduce a method that could be used to determine which task would be most beneficial to train on given that there are a lot of available instruction tuning datasets. The authors test a continual learning/active learning scenario of adding new tasks continuously. Moreover, the authors also propose a TaskMap that could be used to understand the difficulty of a task based on the uncertainty. While the previous DataMap was only used for tasks based on a single-task fine-tuned LM, TaskMap could be used on a variety of tasks with a single instruction tuned LM.

**Questions For The Authors:**

Q1) During perturbation, how could you ensure that the instruction preserve their meaning? For instance, when the word "not" is chosen, it could totally change the meaning.

Q2) It is questionable whether in this scenario, the performance of the already seen tasks are preserved. Does any forgetting happen? If so, would adding a replay buffer be the most viable solution while choosing new tasks to learn with Prompt Uncertainty? This is my biggest concern & question of this paper.

Q3) Would the results hold with even larger dataset scale (e.g., Flan Collection, ShareGPT)?

**Reasons To Accept:**

1. The experiments are solid, tested on diverse settings with multiple runs.
2. The authors suggest a practical solution of how to efficiently & continually learn new tasks.
3. The TaskMap is very practical and could be used in future work.
4. The proposed Prompt Uncetainty method is easy to implement and could be used in multiple scenarios.

**Reasons To Reject:**

1. I do have some concerns regarding whether this method could also preserve the seen task performance (See Question 2)

**Reproducibility:**

5: Could easily reproduce the results.

**Reviewer Confidence:**

4: Quite sure. I tried to check the important points carefully. It's unlikely, though conceivable, that I missed something that should affect my ratings.

---

> ### Author Rebuttal · Authors · 2023-08-29
>
> We thank the reviewer for the constructive feedback.
>
> **[Q1] How to make sure the prompt perturbation preserve the meaning**:
> It is difficult to 100% guarantee the preservation of instruction meaning after automatic paraphrasing methods. To ensure the prompt uncertainty is not measured using an extreme perturbation case, we perturbed all instructions 20 times in our experiments. We also tried several instruction perturbation methods at our early experiment stage, such as randomly repeating tokens or adding extraneous tokens, which achieved similar prompt uncertainty scores as randomly dropping words.
> Additionally, for the NIV2 and Self-Instruct datasets we used, which have detailed instructions with many redundant tokens (average 56 words per instruction), randomly dropping 20% of tokens will mostly preserve the meaning of the instructions.
> For other datasets with concise instructions, a higher dropping rate is needed to perturb the instructions, leading to a higher probability of changing instructions meaning entirely. We will add more discussions in the limitation section for the final version.
>
> **[Q2] Will the method forget seen tasks**:
> We clarify that our setting re-trains a model at every active instruction tuning iteration without applying continual learning, and thus, there is no concern about forgetting seen tasks.
> We do not incorporate continual learning to investigate historical task performance but focus only on generalization performance, which better aligns with the goal of instruction tuning.
> In terms of efficiency, we acknowledge the limitation of this strategy in our limitation section (lines 585 - 588).
> We leave the combination of continual learning and active instruction tuning as an exciting future work.
>
> **[Q3] Will the results hold for larger datasets**:
> Our work focuses on proposing active instruction tuning and prompt uncertainty with a comparable and reproducible setting for future work to further research. We test our method on NIV2 and self-instruct datasets, which have sizes comparable to FLAN and ShareGPT datasets.
>
> **Missing References**:
> Thanks for suggesting these references. We will add these work to related work for the final version.

---

### Official Review · Reviewer_daei · 2023-08-15

**Soundness:** 3

**Excitement:**

3: Ambivalent: It has merits (e.g., it reports state-of-the-art results, the idea is nice), but there are key weaknesses (e.g., it describes incremental work), and it can significantly benefit from another round of revision. However, I won't object to accepting it if my co-reviewers champion it.

**Paper Topic And Main Contributions:**

The study presents a novel framework called Active Instruction Tuning. This approach is designed to effectively enhance the generalizing capacity of Instruction Tuning (IT) models, especially when dealing with expansive instruction tuning scenarios.

The researchers have introduced a unique metric named "Prompt Uncertainty" at the task level for Instruction Tuning. This metric aims to pinpoint new or insightful tasks that can significantly elevate the performance of IT models in unfamiliar or zero-shot scenarios.

Additionally, the team has developed a diagnostic tool called "Task Map." This tool is tailored to categorize tasks based on two criteria: the uncertainty in their prompts and the likelihood of their predictions. By doing so, it offers valuable insights into the nature and quality of different tasks.

**Reasons To Accept:**

The presented work introduces a novel framework, Active Instruction Tuning with prompt uncertainty, aimed at enhancing the generalization capability of Instruction Tuned (IT) models in large-scale instruction tuning scenarios. The main reasons to accept this work are:

Novelty: The proposed method of employing prompt uncertainty as a criterion for task selection is novel and addresses the need for efficient task selection in large-scale instruction tuning settings.

Thorough Experiments: Experiments conducted on two established datasets (NIV2 and Self-Instruct) validate the superiority of the proposed method over other baseline task selection techniques. This provides empirical evidence of the method's effectiveness.

Task Map Tool: The introduction of the Task Map tool, which categorizes tasks based on prompt uncertainty and prediction probability, is a significant contribution. It offers insights into the nature of tasks and their potential impact on model performance, suggesting directions for future research.

Broad Implications: The study's findings have implications beyond just the tested datasets. By revealing that certain "difficult" tasks offer no benefit in instruction tuning, the research prompts a deeper investigation into the nature and quality of tasks used for instruction tuning.

Comprehensive Discussion: The work not only presents experimental results but also discusses the limitations, offering a balanced perspective. This self-awareness suggests areas of improvement and highlights potential pitfalls.

Relevance to Current Research: The research ties into current trends in the machine learning community, like the importance of active learning, uncertainty estimation, and the growth of instruction tuning paradigms, making it timely and relevant.

**Reasons To Reject:**

Scalability: The method is designed to work when the number of tasks is large and continuously expanding. However, how it scales with an ever-increasing number of tasks in terms of efficiency and computational costs is not immediately clear.

Generalizability: The experiments seem to be focused on specific datasets (NIV2 and Self-Instruct). How the method performs on different types of datasets or real-world scenarios is a concern and what scenarios do the current selection of the datasets not cover needs to be discussed further.

Implicit Assumptions: The method is based on certain hypotheses, such as the association between prompt uncertainty and the model's lack of knowledge about a particular task. If these hypotheses are incorrect, then the foundational premise of the method is weakened.

Experimental Weakness: The experiments were conducted only on open-source instruction tuning models. As a result, the generalizability of the findings to more advanced models, especially those employing reinforcement learning with human feedback (as in InstructGPT by Ouyang et al., 2022), remains untested.

The research did not incorporate continual learning in its experiments. Given that continual learning could be a significant factor in realistic settings, especially when dealing with a constantly evolving task pool, the absence of this component might limit the applicability and adaptability of the proposed approach.

**Reproducibility:**

3: Could reproduce the results with some difficulty. The settings of parameters are underspecified or subjectively determined; the training/evaluation data are not widely available.

**Reviewer Confidence:**

5: Positive that my evaluation is correct. I read the paper very carefully and I am very familiar with related work.

---

> ### Author Rebuttal · Authors · 2023-08-29
>
> We thank the reviewer for the constructive feedback.
>
> **Scalability**:
> The computation cost of measuring prompt uncertainty will scale linearly with the number of new tasks. We will include additional details about the scalability of active instruction tuning and prompt uncertainty in the updated version.
>
> **Generalizability**:
> We believe NIV2 and Self-Instruct datasets are two representative datasets for two different types of instruction tuning, described in lines 256 - 265. While the NIV2 dataset is human-curated and has clear task boundaries like FLAN and PromptSource datasets, the Self-Instruct dataset is GPT-generated and does not have clear task boundaries, similar to the ShareGPT dataset used in Vicuna training. We conduct experiments on these two different types of datasets to showcase that our method is generalizable in different scenarios.
> We acknowledge that the current selection of datasets does not contain data from non-English and specific domains (medical, science) in the limitation Section of the final version.
>
> **Implicit Assumptions**:
> Our empirical results show that the proposed prompt uncertainty is associated with task usefulness during instruction tuning.
>
> **Experimental Weakness**:
> We agree that testing on a larger scale and closed-source models might bring more insights, and also acknowledge this limitation at line 576 - 580. However, we do not have access to those closed-sourced models. While Instruct-GPT can be finetuned with OpenAI API, we cannot access the prediction logits and calculate the prompt uncertainty for these closed-source models.
>
> **The research did not incorporate continual learning**:
> While continual learning (CL) centers on preserving performance on historical tasks, our proposed active instruction tuning concentrates on enhancing generalization performance, which better aligns with the objective of instruction tuning. The retraining of new models during iterations of active instruction tuning facilitates a direct evaluation of the selected task utility and the efficacy of task selection methods.
> Regarding efficiency in an application setting, incorporating continual learning could potentially yield further improvements, which we have stated in the limitations section (lines 585 - 588). We leave the combination of continual learning and active instruction tuning as an exciting future work.

---

### Meta-Review · Area_Chair_UPr4 · 2023-09-15

**Recommendation:** 4

**Metareview:**

This work proposes a new metric for identifying tasks for improving active instructing tuning. All reviews agree on the novelty of the proposed metric and usefulness of the study (for example task maps). Authors evaluate their method on large open-source LMs and two different instruction tuning dataset. Reviews also highlight some interesting follow-up questions and future research.

---

### Decision · Program_Chairs · 2023-10-07

**Decision:**

Accept-Main

**Comment:**

This work proposes a new metric for identifying tasks for improving active instructing tuning. All reviews agree on the novelty of the proposed metric and usefulness of the study (for example task maps). Authors evaluate their method on large open-source LMs and two different instruction tuning dataset. Reviews also highlight some interesting follow-up questions and future research.